# Influence of Seasonal Water Level Fluctuations on Food Web Structure of a Large Floodplain Lake in China

**Huan Zhang [1], Yuyu Wang [2] and Jun Xu [3],***

1   School of Life Sciences, Nanchang University, Nanchang 330031, China; huanzhang@ncu.edu.cn
2   School of Ecology and Nature Conservation, Beijing Forestry University, Beijing 100083, China; wangyy@bjfu.edu.cn
3   Institute of Hydrobiology, Chinese Academy of Sciences, Wuhan 430072, China
*   Correspondence: xujun@ihb.ac.cn; Tel.: +86-27-6878-0195

**Abstract:** Seasonal shifts in hydrology are known to alter the abundance and diversity of basal production resources and habitats and hence strongly influence the structure and function of river ecosystems. However, equivalent knowledge of natural lake ecosystems in floodplain regions is lacking. Here, we used stable isotope ratios of carbon and nitrogen to assess available primary production sources and consumer taxa during the dry and wet seasons in a large floodplain lake connected to the Yangtze River. Fish species showed distinct $\delta^{13}$C values between two hydrological periods but only small changes in $\delta^{15}$N values. Most of the fish species had higher estimated trophic levels in the dry season, likely indicating greater carnivory. Results of Bayesian mixing models revealed that benthic algae and benthic organic matter (BOM), combined with $C_3$ vegetation, were the principal food sources supporting the biomass of most fish species during the low-water period, whereas benthic algae and seston were the most important carbon sources during the flood period. Overall, these findings demonstrate that seasonal hydrological changes, such as water-level fluctuations, can affect the trophic structure and ecosystem functioning of floodplain lake food webs in the subtropical zone.

**Keywords:** water-level fluctuations; food web; stable isotopes; trophic position; Lake Dongting





## 1. Introduction

Seasonal change in water level is a critical factor that influences the structure and functioning of freshwater ecosystems [1–3]. Particularly in floodplains, the integrity of processes is closely associated with the hydrological regime, and the alteration between flooding and drought periods determines the structure and functioning of these environments [4,5]. Previous studies have shown that seasonal water-level fluctuations can influence the dynamics and structure of the phytoplankton and zooplankton communities [6,7]. In the littoral zone, water-level fluctuations play a crucial role in the distribution and composition of macrophytes [8] and macroinvertebrates [9] and have an indirect effect on fish community structure [10], ultimately causing variation in trophic interactions of aquatic ecosystems.

Carbon sources available for fish are expected to be scarcer and less variable during periods of low water, whereas flooding during high-water periods brings fish into contact with a greater abundance and diversity of food resources such as terrestrial plants and biofilms that grow on submerged terrestrial plants [11–13]. The gradual drying out of the floodplain, the associated reduction in habitat area and the availability of resources causes fish density to increase, and species interactions to intensify [3,11,14]. Therefore, the trophic relationships of fish may change with the hydrological cycle. On the other hand, seasonal fluctuations in the water level can provoke complicated changes in the community structure of the littoral area [9]. Most of the research concerning the response of aquatic systems to extreme water level fluctuation (i.e., flooding and drought) has taken place in arid climates [15] or focuses primarily on riverine systems [5,16,17]. Little attention has been

paid to the ecological effects of water level fluctuations in naturally fluctuating floodplain lakes in subtropics [18]. Such knowledge will greatly improve our understanding of the responses of aquatic organisms such as fish to hydrological change in lakes. Fluctuations in water level cause variation in connectivity between the main lake area and the floodplain and have the potential to significantly influence food-web structure by changes in potential food sources and movement of consumer taxa [11,14]. Therefore, the hydrodynamics of large floodplain lake ecosystems must be integrated into food web studies to provide more accurate and holistic estimates of carbon sources supporting aquatic food webs at spatial and temporal scales relevant to these systems.

We used stable isotope signatures of primary production sources, macroinvertebrates, and fish to evaluate the food web structure in East Lake Dongting, China, during the dry and wet seasons. East Lake Dongting is one part of the basin of Lake Dongting, which still connects with the Yangtze River, and provides a unique opportunity to examine the food-web structure in a large subtropical floodplain lake. The primary goals of our study were (1) to identify the principal carbon sources supporting consumer taxa in East Lake Dongting during the dry and wet seasons and (2) to compare trophic position of fish species between seasons. To address these questions, the relative contribution of different basal carbon sources to fish assemblage was assessed through Bayesian stable isotope mixing models.

## 2. Materials and Methods

### 2.1. Study Area

Lake Dongting (N 28°40′~29°25′, E 112°00′~113°15′) is the second largest freshwater lake in China, with a surface area of around 3000 km². It is located in the middle reach of the Yangtze River and is mainly divided into three large basins: East Lake Dongting, South Lake Dongting, and West Lake Dongting. Our study was carried out in the eastern basin of Lake Dongting (Figure 1). The lake is affected by seasonal inflows from the major river system during the flooding season and drains back into the Yangtze River when the river level is low. Due to its proximity to the outflow of Lake Dongting, this region of the lake has dramatic seasonal fluctuations in water level, averaging more than 10 m, supporting little aquatic macrophyte growth.

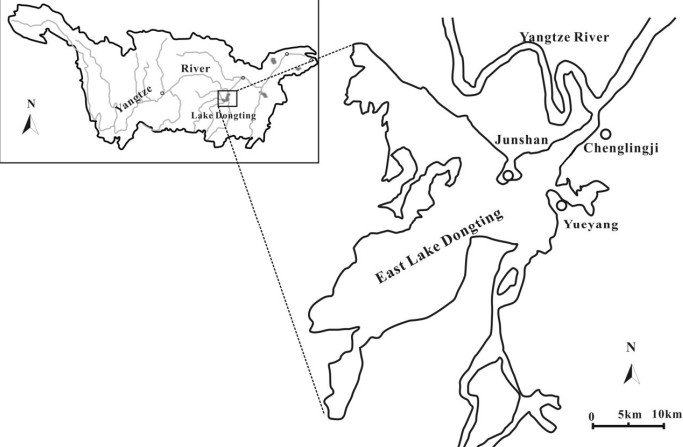

**Figure 1.** Map of the study area in the eastern basin of Lake Dongting in the middle and lower reaches of the Yangtze River basin.

### 2.2. Sample Collection

Isotopic samples collection was conducted during the dry (January to March) and wet (July to September) seasons of 2016. Fish samples were purchased from a local fisherman who caught with seines nets, gill nets, and electric fishing. Species collected for isotopic analysis were selected based on previous surveys that identified them as dominant consumers. We sampled adult individuals of fifteen important fish species with different



feeding guilds: the piscivorous species *Siniperca chuatsi* (*n* = 16); the three carnivorous species *Coilia brachygnathus* (*n* = 22), *Culterichthys erythropterus* (*n* = 11), and *Silurus asotus* (*n* = 10); the five omnivorous species *Hemiculter leucisculus* (*n* = 8), *Xenocypris davidi* (*n* = 8), *Acheilognathus macropterus* (*n* = 16), *Cyprinus carpio* (*n* = 10) and *Carassius auratus* (*n* = 12); and the six invertivorous species *Sarcocheilichthys sinensis* (*n* = 13), *Coreius heterodon* (*n* = 14), *Saurogobio dabryi* (*n* = 13), *Pelteobagrus fulvidraco* (*n* = 10), *P. nitidus* (*n* = 6) and *P. eupogon* (*n* = 11). Meanwhile, two macroinvertebrate species (*Corbisula fluminea* (*n* = 15) and *Bellamya aeruginosa* (*n* = 12)) were also collected using a benthic net.

Water samples were passed through a 64 μm mesh sieve to remove zooplankton and then filtered onto pre-combusted glass fiber filters (Whatman GF/F) following Zeug and Winemiller (2008) [19]; this source is referred to as "seston". Benthic organic matter (BOM) samples were also collected using a weighted Peterson sampler in the littoral habitat (<3 m depth). Additionally, samples of dominant vegetation (C$_3$ species) such as *Carex* sp. and *Phalaris arundinacea* were collected from adjacent riparian areas when their leaves were available during flood and dry seasons. Fresh leaf samples were clipped, placed in plastic bags, and frozen for later processing. Samples of filamentous algae were scraped from substrata (i.e., rocks), then rinsed with distilled water to remove sediment and large particles of detritus. As this collection was unlikely to yield pure samples, this basal production source, composed of attached filamentous algae and probably other materials is referred to as "benthic algae".

All samples were placed in plastic bags and frozen for later processing. Muscle tissue was dissected from the dorsal musculature of adult fish samples. Meanwhile, samples of abdominal muscle were used for macroinvertebrates. Muscle and processed basal source material samples were then placed in individually labelled, acid-washed Petri dishes and dried for 48 h at 60 °C. Dried samples were ground to a fine powder using a pestle and mortar.

### 2.3. Stable Isotope Analysis

Measurements of stable carbon and nitrogen isotope ratios were performed with an EA 1110 elemental analyzer (Carlo Erba) coupled to a Finnigan Delta Plus continuous flow isotope ratio mass spectrometer (Thermo Scientific, Waltham, MA, USA). Isotopic ratios were expressed in delta (δ) notation per thousand (‰) according to the following equation: $\delta X(‰) = [(R_{sample}/R_{standard}) - 1] \times 1000$, where $R = {}^{13}C/{}^{12}C$ or $R = {}^{15}N/{}^{14}N$. The standards for $\delta^{13}C$ and $\delta^{15}N$ were Vienna Pee Dee Belemnite (PDB) and atmospheric N$_2$, respectively [20]. About 20% of the samples were analyzed using two or more replicates. Two standards were also run after every 10 samples to compensate for the drift over time. The analytical deviations in $\delta^{13}C$ and $\delta^{15}N$ replicate analyses were both within 0.3‰.

### 2.4. Data Analysis and Food Source Modelling

We examined differences in the base of food webs between the dry and wet seasons in Lake Dongting by estimating the proportional contribution of basal sources to consumer diets using stable isotope mixing models in R with simmr [21]. The basal resources considered in models for consumers were seston, BOM, benthic algae, and C$_3$ vegetation during both seasons. Prior to analysis, we evaluated the data to ensure it conformed to the mixing model assumptions. One assumption is that consumer isotopic values fall within the polygon defined by C and N isotopic values of potential food sources when they are adjusted to account for trophic fractionation [22]. Carbon and nitrogen values were corrected for trophic fractionation using the respective values of 0.4‰ ± 1.3‰, and 3.4‰ ± 1.0‰ [23]. Trophic position was determined using a standard two-source mixing model [23]: Trophic position = $\lambda + (\delta^{15}N_{consumer} - [\delta^{15}N_{base1} \times \alpha + \delta^{15}N_{base2} \times (1 - \alpha)])/3.4$, where λ was the trophic level of food bases (e.g., 2 for primary consumers); $\delta^{15}N_{consumer}$ was the $\delta^{15}N$ for consumer; and 3.4‰ was the average nitrogen isotope fractionation per trophic level. Trophic position calculations used a baseline based on mussels, which mostly reflect the isotopic signature of seston that forms the base of the pelagic food web, and snails, which



tend to reflect the isotopic signature of detritus and periphyton from the littoral food web [23].

To compare the relative contributions of littoral and pelagic carbon sources to the fish species in the two seasons, a two-source mixing model was used to estimate the contribution of littoral secondary production to fishes using the formula:

$$\text{Percentage contribution of littoral} = (\delta^{13}C_c - \delta^{13}C_p)/(\delta^{13}C_l - \delta^{13}C_p)$$

where, $\delta^{13}C_c$, $\delta^{13}C_p$, and $\delta^{13}C_l$ are $\delta^{13}C$ values of fish species, pelagic baseline (mean $\delta^{13}C$ of *C. fluminea*), and littoral baseline (mean $\delta^{13}C$ of *B. aeruginosa*), respectively [23,24].

Then, we used analysis of variance to test for seasonal differences in C and N isotope signatures of primary production sources and fish consumers between the dry and flood periods. Meanwhile, trophic positions of fishes were also compared by one-way ANOVA. Non-metric multidimensional scaling (NMDS) was conducted to examine the differences of basal production source contributions to fish species between hydrological seasons.

## 3. Results

### 3.1. $\delta^{13}C$ and $\delta^{15}N$ Signatures in Basal Food Resources

Mean $\delta^{13}C$ values of basal food sources showed seasonal differences and could be clearly distinguished (Tables 1 and 2). In the dry season, $\delta^{13}C$ values of basal food sources ranged from $-29.8‰$ to $-21.3‰$, and in the wet season from $-27.8‰$ to $-20.5‰$. Seston had significantly higher $\delta^{13}C$ values during the dry season ($F = 30.9$, $p < 0.01$), but benthic algae and $C_3$ vegetation were markedly more $^{13}C$-enriched in the flood period ($F = 12.3$, $p < 0.05$ for benthic algae; $F = 38.85$, $p < 0.01$ for $C_3$ vegetation).

**Table 1.** Sample sizes and mean ($\pm$ SD) $\delta^{13}C$ and $\delta^{15}N$ values (‰) for consumer taxa collected in Lake Dongting during the dry and wet seasons.

| Species | Abbreviation | Dry Season | | | Wet Season | | |
|---|---|---|---|---|---|---|---|
| | | N | $\delta^{13}C$ (‰) | $\delta^{15}N$ (‰) | N | $\delta^{13}C$ (‰) | $\delta^{15}N$ (‰) |
| Macroinvertebrate | | | | | | | |
| *Corbisula fluminea* | Cfl | 10 | $-26.3 \pm 0.3$ | $6.9 \pm 0.5$ | 5 | $-28.5 \pm 0.4$ | $7.1 \pm 0.5$ |
| *Bellamya aeruginosa* | Bae | 6 | $-24.5 \pm 0.3$ | $5.5 \pm 0.6$ | 6 | $-26.9 \pm 0.8$ | $6.9 \pm 0.30$ |
| Fish | | | | | | | |
| *Coilia brachygnathus* | Cbr | 12 | $-25.8 \pm 1.9$ | $14.6 \pm 1.1$ | 10 | $-27.2 \pm 1.3$ | $12.5 \pm 0.7$ |
| *Hemiculter leucisculus* | Hle | 3 | $-25.9 \pm 0.3$ | $8.8 \pm 1.0$ | 5 | $-28.6 \pm 0.6$ | $10.5 \pm 0.8$ |
| *Culterichthys erythropterus* | Cer | 4 | $-26.1 \pm 0.3$ | $12.6 \pm 0.8$ | 7 | $-25.7 \pm 0.4$ | $11.4 \pm 0.7$ |
| *Xenocypris davidi* | Xda | 3 | $-24.0 \pm 0.5$ | $9.1 \pm 0.9$ | 5 | $-24.9 \pm 1.0$ | $8.6 \pm 0.2$ |
| *Sarcocheilichthys sinensis* | Ssi | 6 | $-25.6 \pm 1.5$ | $10.8 \pm 1.0$ | 7 | $-25.4 \pm 1.3$ | $10.1 \pm 1.0$ |
| *Coreius heterodon* | Che | 8 | $-24.9 \pm 0.8$ | $11.7 \pm 1.1$ | 6 | $-26.8 \pm 0.5$ | $12.2 \pm 0.4$ |
| *Saurogobio dabryi* | Sda | 7 | $-23.4 \pm 0.6$ | $10.4 \pm 1.0$ | 7 | $-26.2 \pm 0.5$ | $12.0 \pm 0.3$ |
| *Acheilognathus macropterus* | Ama | 9 | $-24.3 \pm 1.0$ | $8.9 \pm 0.8$ | 7 | $-24.2 \pm 1.5$ | $9.4 \pm 1.2$ |
| *Cyprinus carpio* | Cca | 5 | $-25.1 \pm 0.2$ | $11.4 \pm 1.4$ | 5 | $-25.7 \pm 0.6$ | $11.0 \pm 0.8$ |
| *Carassius auratus* | Cau | 8 | $-24.2 \pm 0.9$ | $9.8 \pm 1.4$ | 4 | $-25.2 \pm 0.7$ | $10.3 \pm 1.5$ |
| *Silurus asotus* | Sas | 4 | $-23.4 \pm 1.0$ | $12.0 \pm 0.3$ | 6 | $-24.7 \pm 0.3$ | $11.8 \pm 1.0$ |
| *Pelteobagrus fulvidraco* | Pfu | 5 | $-23.3 \pm 1.2$ | $10.1 \pm 1.1$ | 5 | $-23.5 \pm 1.2$ | $9.8 \pm 1.6$ |
| *Pelteobagrus nitidus* | Pni | 3 | $-24.7 \pm 0.5$ | $10.4 \pm 1.4$ | 3 | $-26.5 \pm 0.7$ | $12.1 \pm 0.8$ |
| *Pelteobagrus eupogon* | Peu | 8 | $-25.4 \pm 0.8$ | $9.9 \pm 1.0$ | 3 | $-24.2 \pm 0.7$ | $10.3 \pm 0.8$ |
| *Siniperca chuatsi* | Sch | 8 | $-23.6 \pm 0.3$ | $12.8 \pm 0.7$ | 8 | $-24.5 \pm 0.9$ | $11.2 \pm 0.8$ |

**Table 2.** ANOVA analysis for isotope values ($\delta^{13}$C and $\delta^{15}$N) and trophic level (TL) of fish consumers collected in Lake Dongting between the dry and wet seasons.

| Species | $\delta^{13}$C (‰) | | | $\delta^{15}$N (‰) | | | TL | | |
|---|---|---|---|---|---|---|---|---|---|
| | df | F | p | df | F | p | df | F | p |
| *Coilia brachygnathus* | 1 | 3.754 | 0.067 | 1 | 28.593 | **<0.001** | 1 | 94.07 | **<0.01** |
| *Hemiculter leucisculus* | 1 | 47.608 | **<0.001** | 1 | 6.813 | **0.04** | 1 | 0.163 | 0.700 |
| *Culterichthys erythropterus* | 1 | 4.103 | 0.074 | 1 | 7.183 | **0.025** | 1 | 24.950 | **<0.01** |
| *Xenocypris davidi* | 1 | 1.309 | 0.296 | 1 | 1.654 | 0.246 | 1 | 56.94 | **<0.01** |
| *Sarcocheilichthys sinensis* | 1 | 0.046 | 0.835 | 1 | 1.538 | 0.241 | 1 | 24.31 | **<0.01** |
| *Coreius heterodon* | 1 | 24.629 | **<0.01** | 1 | 0.318 | 0.583 | 1 | 5.558 | **0.036** |
| *Saurogobio dabryi* | 1 | 95.800 | **<0.01** | 1 | 15.7 | **<0.01** | 1 | 6.737 | **0.023** |
| *Acheilognathus macropterus* | 1 | 0.123 | 0.732 | 1 | 1.087 | 0.316 | 1 | 11.12 | **<0.01** |
| *Cyprinus carpio* | 1 | 2.873 | 0.151 | 1 | 60.813 | **<0.01** | 1 | 5.552 | 0.065 |
| *Carassius auratus* | 1 | 4.208 | 0.067 | 1 | 0.238 | 0.636 | 1 | 3.979 | 0.074 |
| *Silurus asotus* | 1 | 9.401 | **0.015** | 1 | 0.061 | 0.811 | 1 | 23.72 | **<0.01** |
| *Pelteobagrus fulvidraco* | 1 | 0.055 | 0.821 | 1 | 0.104 | 0.756 | 1 | 9.887 | **0.014** |
| *Pelteobagrus nitidus* | 1 | 12.768 | **0.023** | 1 | 3.199 | 0.148 | 1 | 0.445 | 0.541 |
| *Pelteobagrus eupogon* | 1 | 5.314 | **0.047** | 1 | 0.268 | 0.617 | 1 | 7.106 | **0.026** |
| *Siniperca chuatsi* | 1 | 7.180 | **0.018** | 1 | 16.544 | **<0.01** | 1 | 141.00 | **<0.01** |

A significant difference in $\delta^{15}$N signatures between seasons was only apparent for BOM, which had higher $\delta^{15}$N values during the wet season ($F = 8.982$, $p < 0.05$). $C_3$ vegetation was more $^{15}$N-depleted in the low-water period, while seston and benthic algae both had similar $\delta^{15}$N values during the dry and wet seasons.

### 3.2. $\delta^{13}$C and $\delta^{15}$N Values in Consumers

Consumer taxa in Lake Dongting exhibited significant seasonal changes in $\delta^{13}$C values (Table 1; Figure 2). $\delta^{13}$C signatures of macroinvertebrates (*C. fluminea* and *B. aeruginosa*) were significantly lower in the wet season (*C. fluminea*: $F = 626.123$, $p < 0.01$; *B. aeruginosa*: $F = 150.758$, $p < 0.01$). During the low-water period, most of the fish species were $^{13}$C-enriched; however, 8 of 15 species exhibited little differences in $\delta^{13}$C values between hydrological seasons (Table 2).

There were no temporal variations in consumer $\delta^{15}$N signatures between the dry and wet seasons, although most of the fish species had higher $\delta^{15}$N values during the low-water period (Table 1, Figure 2). *C. brachygnathus* and *C. erythropterus* exhibited highest $\delta^{15}$N values during the dry season when compared with other fish species; however, *C. erythropterus* were $^{15}$N-depleted relative to other predators in the wet season. Additionally, macroinvertebrates (*C. fluminea* and *B. aeruginosa*) were more $^{15}$N-enriched in the wet season.

### 3.3. Trophic Level of Fish Species

Mean estimated trophic level of fishes showed obvious differences between the dry and wet seasons ($F = 77.33$, $p < 0.01$, Table 2), with most species showing higher trophic levels in the dry season (Figure 3). Only 4 of 15 fish species were found to have no significant seasonal variation in trophic level (Table 2). In the dry season, the mean trophic level by species ranged from 2.67 for *H. leucisculus* to 4.35 for *C. brachygnathus*, whereas the wet season range was 2.18 for *X. davidi* to 3.31 for *C. brachygnathus* and 3.24 for *C. heterodon*.

### 3.4. Carbon Sources Supporting Aquatic Consumers

Mean relative contributions of basal food sources to fish species exhibited many differences between the two hydrological seasons (Tables 3 and 4). The NMDS ordination verified these results in Lake Dongting (Figure 4). Benthic algae had consistently high contribution to most fish species in Lake Dongting in both seasons, with mean feasible contributions ranging from 25% to 46% during the dry season and from 18% to 53% during the wet season, respectively (Tables 3 and 4). Results from mixing models suggested that fish depended on benthic algae and seston when water level was high, indicating their importance to the biomass of consumers. However, seston did not appear to be an important food source to fish consumers during the low-water period.

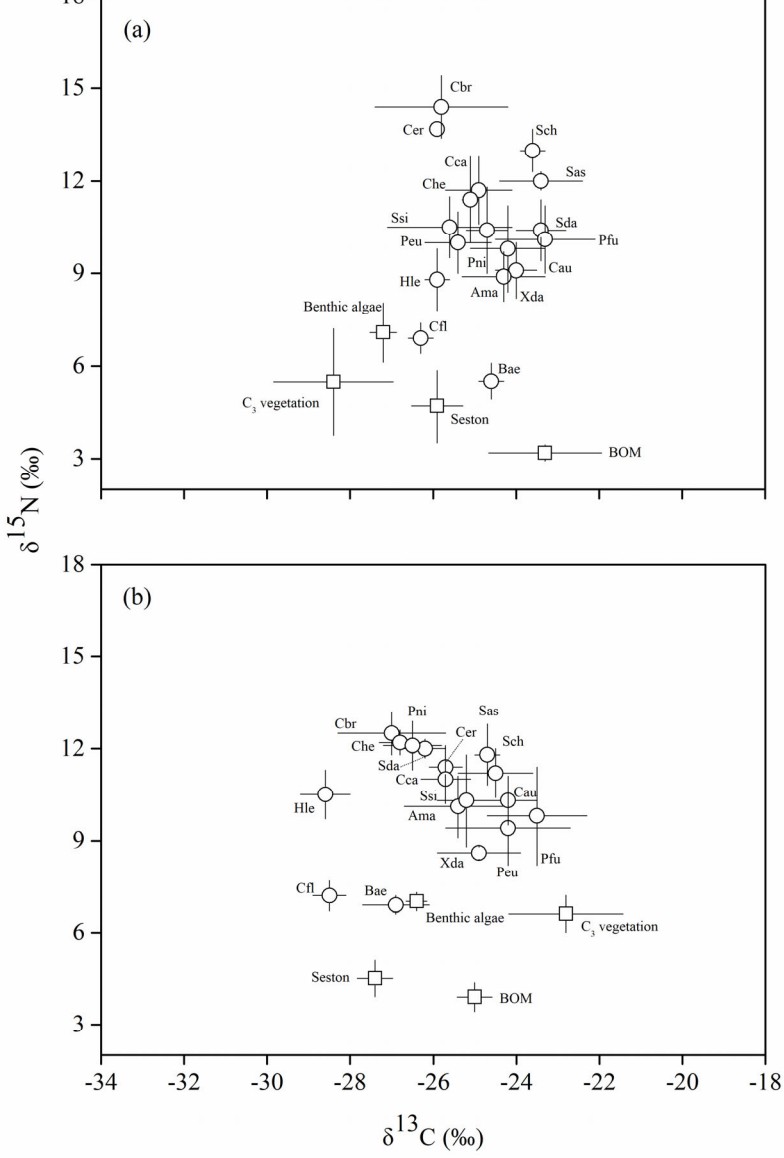

**Figure 2.** $\delta^{13}C$ and $\delta^{15}N$ values (mean ± SD) of basal food sources (squares), macroinvertebrate, and fish consumers (circles) in Lake Dongting during the dry (**a**) and wet (**b**) seasons. Consumers: Cfl, *Corbisula fluminea*; Bae, *Bellamya aeruginosa*; Cbr, *Coilia brachygnathus*; Hle, *Hemiculter leucisculus*; Cer, *Culterichthys erythropterus*; Xda, *Xenocypris davidi*; Ssi, *Sarcocheilichthys sinensis*; Che, *Coreius heterodon*; Sda, *Saurogobio dabryi*; Ama, *Acheilognathus macropterus*; Cca, *Cyprinus carpio*; Cau, *Carassius auratus*; Sas, *Silurus asotus*; Pfu, *Pelteobagrus fulvidraco*; Pni, *Pelteobagrus nitidus*; Peu, *Pelteobagrus eupogon*; Sch, *Siniperca chuatsi*.

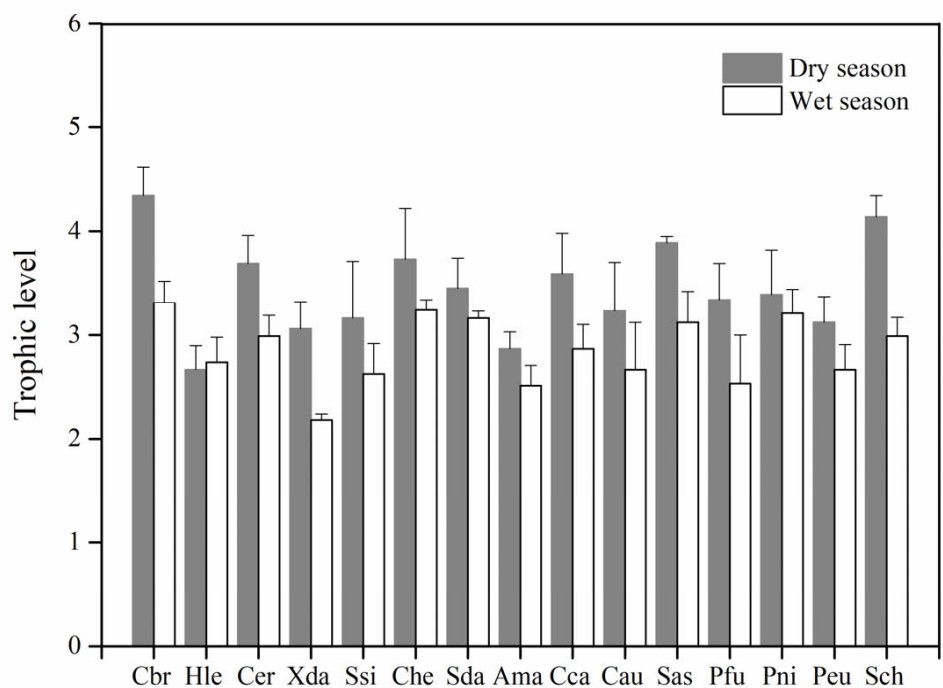

**Figure 3.** Mean trophic level estimates for fishes in Lake Dongting between the dry (grey) and wet (white) seasons. Cbr, *Coilia brachygnathus*; Hle, *Hemiculter leucisculus*; Cer, *Culterichthys erythropterus*; Xda, *Xenocypris davidi*; Ssi, *Sarcocheilichthys sinensis*; Che, *Coreius heterodon*; Sda, *Saurogobio dabryi*; Ama, *Acheilognathus macropterus*; Cca, *Cyprinus carpio*; Cau, *Carassius auratus*; Sas, *Silurus asotus*; Pfu, *Pelteobagrus fulvidraco*; Pni, *Pelteobagrus nitidus*; Peu, *Pelteobagrus eupogon*; Sch, *Siniperca chuatsi*.

**Table 3.** Means and low 95% and high 95% credit percentile ranges (in parentheses) of source contributions to fish biomass during the dry season. Numbers in bold indicate assimilation of that source (1st-percentile values greater than 0).

| Species | Seston | BOM | Benthic Algae | C$_3$ Vegetation |
|---|---|---|---|---|
| *Coilia brachygnathus* | 0.19 (0.00–0.42) | 0.09 (0.00–0.24) | **0.41 (0.05–0.78)** | **0.31 (0.01–0.57)** |
| *Hemiculter leucisculus* | 0.24 (0.00–0.47) | 0.14 (0.00–0.33) | **0.28 (0.01–0.51)** | **0.34 (0.06–0.58)** |
| *Culterichthys erythropterus* | 0.18 (0.00–0.41) | 0.07 (0.00–0.22) | **0.37 (0.02–0.68)** | **0.37 (0.08–0.63)** |
| *Xenocypris davidi* | 0.26 (0.00–0.49) | **0.28 (0.07–0.47)** | **0.28 (0.02–0.49)** | 0.18 (0.00–0.38) |
| *Sarcocheilichthys sinensis* | 0.18 (0.00–0.42) | 0.09 (0.00–0.25) | **0.46 (0.11–0.87)** | 0.27 (0.00–0.52) |
| *Coreius heterodon* | 0.25 (0.00–0.48) | 0.16 (0.00–0.30) | **0.36 (0.03–0.69)** | 0.24 (0.00–0.43) |
| *Saurogobio dabryi* | 0.24 (0.00–0.48) | **0.39 (0.19–0.59)** | **0.28 (0.01–0.52)** | 0.09 (0.00–0.26) |
| *Acheilognathus macropterus* | 0.25 (0.00–0.48) | **0.27 (0.06–0.45)** | **0.27 (0.05–0.47)** | 0.21 (0.00–0.41) |
| *Cyprinus carpio* | 0.25 (0.00–0.47) | 0.11 (0.00–0.24) | **0.41 (0.11–0.75)** | **0.23 (0.01–0.42)** |
| *Carassius auratus* | 0.24 (0.00–0.49) | **0.27 (0.07–0.44)** | **0.33 (0.04–0.60)** | 0.16 (0.00–0.35) |
| *Silurus asotus* | 0.25 (0.00–0.48) | **0.33 (0.03–0.57)** | 0.25 (0.00–0.49) | 0.17 (0.00–0.39) |
| *Pelteobagrus fulvidraco* | 0.24 (0.00–0.48) | 0.28 (0.00–0.51) | **0.31 (0.01–0.58)** | 0.17 (0.00–0.40) |
| *Pelteobagrus nitidus* | 0.26 (0.00–0.50) | 0.19 (0.00–0.37) | **0.32 (0.01–0.58)** | 0.23 (0.00–0.43) |
| *Pelteobagrus eupogon* | 0.19 (0.00–0.40) | 0.07 (0.00–0.17) | **0.46 (0.18–0.77)** | **0.28 (0.02–0.49)** |
| *Siniperca chuatsi* | 0.27 (0.00–0.51) | **0.38 (0.25–0.53)** | **0.25 (0.01–0.46)** | 0.09 (0.00–0.24) |

**Table 4.** Means and low 95% and high 95% credit percentile ranges (in parentheses) of source contributions to fish biomass during the wet season. Numbers in bold indicate assimilation of that source (1st-percentile values greater than 0).

| Species | Seston | BOM | Benthic Algae | C₃ Vegetation |
|---|---|---|---|---|
| *Coilia brachygnathus* | 0.32 (0.00–0.67) | 0.09 (0.00–0.30) | **0.46 (0.09–0.88)** | 0.13 (0.00–0.36) |
| *Hemiculter leucisculus* | 0.21 (0.00–0.45) | 0.13 (0.00–0.34) | **0.41 (0.06–0.78)** | 0.25 (0.00–0.52) |
| *Culterichthys erythropterus* | **0.42 (0.02–0.50)** | 0.06 (0.00–0.19) | **0.46 (0.12–0.85)** | 0.06 (0.00–0.16) |
| *Xenocypris davidi* | **0.36 (0.12–0.61)** | **0.32 (0.09–0.53)** | 0.18 (0.00–0.34) | 0.14 (0.00–0.31) |
| *Sarcocheilichthys sinensis* | 0.24 (0.00–0.45) | 0.13 (0.00–0.32) | **0.41 (0.10–0.74)** | 0.22 (0.00–0.45) |
| *Coreius heterodon* | 0.36 (0.00–0.76) | 0.13 (0.00–0.34) | 0.36 (0.00–0.70) | 0.15 (0.00–0.39) |
| *Saurogobio dabryi* | **0.49 (0.08–0.85)** | 0.08 (0.00–0.23) | 0.35 (0.00–0.70) | 0.08 (0.00–0.25) |
| *Acheilognathus macropterus* | **0.25 (0.01–0.45)** | **0.22 (0.01–0.42)** | **0.27 (0.01–0.50)** | **0.26 (0.01–0.47)** |
| *Cyprinus carpio* | 0.28 (0.00–0.54) | 0.20 (0.00–0.43) | 0.29 (0.00–0.56) | 0.22 (0.00–0.46) |
| *Carassius auratus* | **0.35 (0.06–0.61)** | 0.15 (0.00–0.36) | **0.35 (0.02–0.63)** | 0.15 (0.00–0.36) |
| *Silurus asotus* | **0.26 (0.02–0.45)** | 0.17 (0.00–0.37) | **0.45 (0.13–0.84)** | 0.12 (0.00–0.28) |
| *Pelteobagrus fulvidraco* | 0.18 (0.00–0.38) | 0.22 (0.00–0.44) | **0.26 (0.01–0.49)** | **0.33 (0.07–0.56)** |
| *Pelteobagrus nitidus* | 0.30 (0.00–0.59) | 0.19 (0.00–0.43) | 0.30 (0.00–0.56) | 0.21 (0.00–0.45) |
| *Pelteobagrus eupogon* | 0.22 (0.00–0.41) | 0.20 (0.00–0.41) | **0.34 (0.02–0.64)** | **0.25 (0.01–0.46)** |
| *Siniperca chuatsi* | 0.17 (0.00–0.38) | 0.09 (0.00–0.28) | **0.53 (0.18–0.91)** | 0.20 (0.00–0.39) |

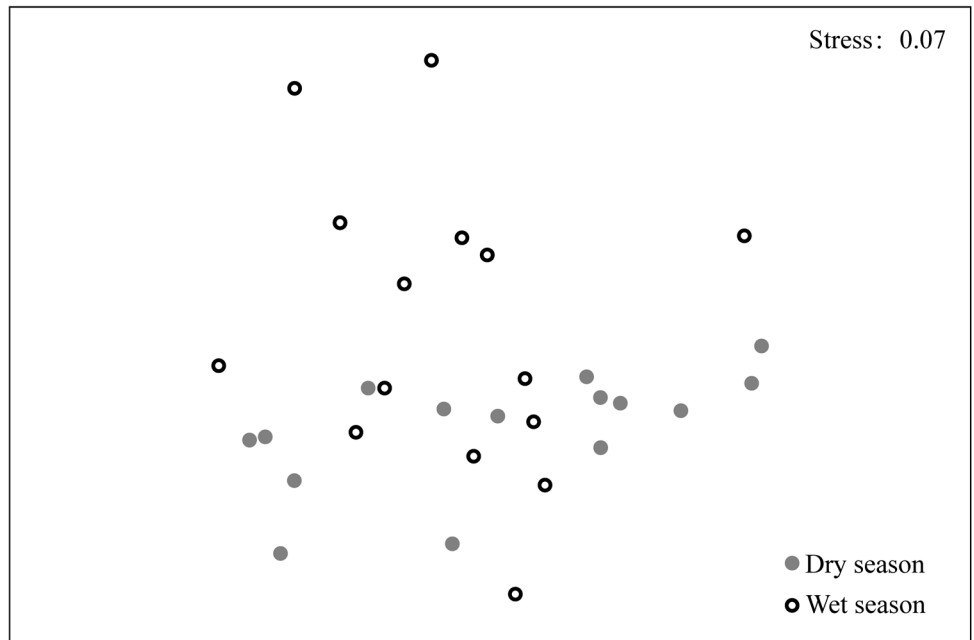

**Figure 4.** Non-metric multidimensional scaling (NMDS) for differences in carbon sources' use of fishes during the dry (grey) and wet (white) seasons in Lake Dongting.

## 4. Discussion

Seasonal changes in water-level regimes are key drivers of the ecosystem structure and functioning of shallow lakes [1,2,13]. Previous stable isotope studies found the food webs of floodplain lakes in the Yangtze basin, such as Poyang Lake and Gucheng Lake, were strongly influenced by changes in the abundance and accessibility of different basal food sources that occur due to hydrological change [3,13,25]. Our results showed that isotopic signatures of consumers and their food sources varied between hydrological

seasons in Lake Dongting. Significant changes in the carbon sources could affect the assemblages of macroinvertebrates and fish through the trophic pathway [26], thereby ultimately influencing the trophic structure of the lake food web, which are reflected in the stable isotope values. Basal carbon sources in Lake Dongting showed variations in isotopic signatures between high and low-water-level periods. Stable isotope ratios of autochthonous primary producers such as phytoplankton and benthic algae can be affected by many environmental factors including water velocity, carbon dioxide concentration, and nutrient levels [27,28]. During the flood season, the water velocity in Lake Dongting is much greater [29], thereby influencing not only the composition and abundance of basal carbon sources but also isotopic values. Similarly, variations in $\delta^{13}$C signatures of benthic algae affected by water velocity can transmit up the food web to consumers [27].

Most of the fish species in our study showed an increase in $\delta^{13}$C signatures from the wet season to the dry season, probably related to available food sources consumed in different seasons. Wantzen et al. [30] proposed that the isotopic shifts of fish in a Pantanal wetland were associated with habitat changes due to a flood pulse, which could be further related to more abundant and variable food sources during the inundation period and increasing carnivory and starvation during the dry season when the lake is reduced. As most of the fish species in Lake Dongting are locally resident [31], differences in fish $\delta^{13}$C signatures between seasons in our study can also reflect seasonal changes in the isotopic values of basal food sources.

In shallow lakes, autochthonous carbon sources, particularly aquatic macrophytes, are typically considered the most important carbon sources supporting consumer growth [16,17,32]. In our study area, the abundance and biomass of aquatic macrophytes are very low due to rapid flow and relatively high turbidity in the wet season [33]. However, aquatic macrophytes seem to not be an important production source to fish, or their contribution was limited to the detrital pathway because of their relative scarcity [34]. In contrast, the biomass of total algae in Lake Dongting was much larger in the flood period than in the dry season [7]. Moreover, benthic algae apparently contributed broadly to 10 of 15 fish species captured from Lake Dongting, being significant for predatory species *C. brachygnathus*, *C. erythropterus,* and *S. chuatsi*, which probably related to their preference for prey that had consumed benthic algae. Winemiller and Jepsen [14] suggested that fish assemblages from regions with predictable flood pulses are able to use highly diverse and quickly changing food items. In that case, a mixture of benthic algae and seston could be the primary energy source that supports consumer taxa during the flood season in Lake Dongting.

The extent to which fish species are affected by water level fluctuations depends on their mobility and habitat requirements [14,35]. In tropical floodplain lakes, fish ingest relatively more vegetative food during the high-water period as fish become more dispersed in the vast flooded area [30,36], whereas an increased carnivory is observed during the dry season when fish become confined to the remaining water body and food resources become limited [30]. Additionally, Cummins [37] suggested that fish feed primarily on benthic macroinvertebrates but not much zooplankton, reflecting consumption of a wide range of both detrital and benthic carbon sources. Recent isotopic evidence further supports that benthic algae and terrestrial riparian vegetation can play important roles in supporting invertebrate consumers and fish in lakes, especially during the low-water period [3,38–40]. According to our model results, fish assemblages in Lake Dongting derived carbon sources especially from benthic algae and seston during the high-water period and relied more on benthic algae, BOM, and $C_3$ vegetation in the dry season.

Our findings indicated that mean trophic positions of most species of fish significantly increased from the wet season to the dry season in Lake Dongting. It is possible that fish feed on prey with higher food quality as fish densities increase, facilitating predator–prey interactions when habitat volume decreases [30,41]. Zhang et al. [26] found that the anchovy, *C. brachygnathus*, consumed more carbon sources from shrimps and small fish in the dry season. Some carnivorous fish species have been shown to feed at lower

trophic levels during high-water periods [42]. During the period of high water level, a large variety of food sources were available for fish, and fish became more dispersal in the vast flooded area, thereby reducing inter- and intraspecific competition. In contrast, Roach et al. [43] suggested that trophic positions of fish assemblages were highest in the flood season in the upper Mississippi River, where additional food items brought by flooding became accessible for fish from terrestrial sources of floodplain, and greater prey diversity may have allowed predators to forage opportunistically on larger prey of a higher trophic position.

However, 4 out of the 15 fish species in our study showed no significant differences in trophic positions between seasons, and these were omnivores, such as *H. leucisculus*, *C. carpio*, *C. auratus*, and *P. nitidus*. Omnivory is considered an adaptation to variability in food resource supply, taking advantage of whatever food is available in a variable and unpredictable environment [44,45]. In floodplain lakes such as Lake Dongting, omnivorous species constitute a major proportion of the fish harvest [10,31]. Due to their strong adaptation and wide utilization of food sources, omnivores would be expected to experience low impacts from naturally fluctuating water levels [44,45], and hence their trophic positions had only weak differences between the dry and wet seasons.

Our results highlight that the food web of floodplain lakes could be altered by seasonally varying water levels associated with changes in the availability of carbon sources and habitats. We found that the food web of East Lake Dongting relied on a mixture of benthic–detrital basal resources in the dry season but switched to pelagic–benthic resources in the wet season. Moreover, seasonal shifts in isotopic signatures of consumers seemed to respond differently to changes in available food sources when hydrological disturbance occurred. The variability in trophic position of fish was associated with their feeding behavior when available habitats were lost or gained, ultimately affecting the trophic interactions in the food webs. These changes present a challenge to untangle the complex trophic relationships in naturally fluctuating lakes such as Lake Dongting. Therefore, understanding the food web dynamics of large floodplain lakes and their response to fluctuating environments such as hydrological seasonality is essential for their management and conservation.

**Author Contributions:** Conceptualization, J.X., Y.W. and H.Z.; Methodology, H.Z. and Y.W.; Writing—original draft, H.Z. and Y.W.; Writing—review and editing, H.Z. and J.X.; supervision, J.X.; funding acquisition, J.X. All authors have read and agreed to the published version of the manuscript.

**Funding:** This work was supported by the National Natural Science Foundation of China (grant No. 31700403).

**Institutional Review Board Statement:** Not applicable.

**Informed Consent Statement:** Not applicable.

**Data Availability Statement:** Data available on request.

**Acknowledgments:** We thank Yuyu Wang for providing some isotopic data and four anonymous reviewers for their helpful suggestions on this manuscript.

**Conflicts of Interest:** The authors declare no conflict of interest.

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
