# Peer review of "Influence of Seasonal Water Level Fluctuations on Food Web Structure of a Large Floodplain Lake in China"

_sustainability, doi:10.3390/su151310724_

Round 1

Reviewer 1 Report

Current study was conducted to determine the variation in food web structure during wet and dry season would be very valuable for the local cosumers and aquaculture research. I agree with its study on seasonal variation affecting on Tropic level of fishes during dry and wet season. All work was good and satisfactory 

Author Response

Reviewer #1:

Current study was conducted to determine the variation in food web structure during wet and dry season would be very valuable for the local consumers and aquaculture research. I agree with its study on seasonal variation affecting tropic level of fishes during the dry and wet seasons. All work was good and satisfactory.

R: We appreciate the referee's kindly comments to us.

Reviewer 2 Report

review report attached

minor changes are suggested

Author Response

Response to Reviewer’s Comments

Reviewer #2:

Specific comments.

1) Title: Seasonal Water-Level Fluctuations Alter the Food Web Structure of a Large Floodplain Lake.

The present title seems very rudimentary so needs to put some more effort to revise the title and it should end with question mark. Change the title to make it more catchy.

R: We appreciate the referee making this recommendation to us. Our manuscript's current title has been changed in accordance with your recommendation (lines 2-3). As a large subtropical floodplain lake, our study sought to examine how does food web structure of Lake Dongting change associated with seasonally hydrological fluctuation. Therefore, even though it looks simple, the former title was chosen in light of our findings. In fact, the title's revision, which ends in a question mark, is more appealing.

2) Abstract

Line 18-19 need to clarify more.

In abstract section a flow is missing so needs to be rephrased.

R: We thank the referee for pointing out this reference to us. Referee 3 made a similar comment (line 18). There are indeed some logical issues in this part, and we apologize for the confusion. We have corrected and reworded the abstract (lines 15-30) to make this part more accurate. Thank you very much for your helpful advice.

3) Introduction

I suggest authors to put 80 % references should be beyond 2018. The old references can be replaced with latest references.

R: We have replaced the old references with the latest ones (lines 393-447) based on your suggestion.

4) Material and methods

ok

R: Thanks you.

5) Results

Discussion and results have a losses coherence so discussion need significant changes as per coherent between results and discussion

R: Thank you for suggestion. Referee 3 made a similar comment. So we have revised and reworded both of the results and discussion parts (lines 193-360) to make their structure more coherent and hope that it is now clearer.

6) Conclusion

There is a need to put a concise and precise conclusion as recommendation of this study.

R: Thank you for your helpful comment. We have reworded the conclusion part (lines 361-378) to make it more precise.

Reviewer 3 Report

Dear Authors,

I appreciate your effort on this interesting research. I have some comments for the manuscript and check whether those are useful for the improvement of the paper.

Thank you

Introduction

Line 49 and 57 – Some suggestions to change the wording. Comments were given in the manuscript attached herewith

Methodology

Scientific names in the section 2.2 under the sub title “Sample collection”, have to be italicized

Line 122, 123 and 149 – check whether, the elaborated symbols/terms have correctly mentioned. Some suggestions were given in the manuscript attached herewith.

Results

Why table 1 has placed after the table 2? it is better if the tables are placed in order.

Figure 4 – Does figure 4 give a meaningful relationship to support the argument related to it? If this relationship created with the 13C signatures and the tropic status (x axis 13C and y axis tropic level for two seasons) it would be more meaningful.

Discussion

Authors have stated “Dongting derived carbon sources especially from benthic algae during the high-water period and relied more on seston and BOM in the dry season” in the discussion probably based on the values in the table 3 and 4. However, there is no strong evidences to prove this argument except the values in the table. If you can do a multivariate analysis for (i.e. PCA or MDS etc.) data in table 3 and 4, the positioning of diffident species in clusters during two seasons will give a good visualization of the changes during two seasons.

Note that in the result section, line 165 and 166, authors have stated that “No significant difference in the δ13C signatures of BOM between seasons was observed (F = 5.231, p = 0.062)”. It seems that these two arguments are conflicting.

Author Response

Response to Reviewer’s Comments

Reviewer #3:

Dear Authors,

I appreciate your effort on this interesting research. I have some comments for the manuscript and check whether those are useful for the improvement of the paper.

Thank you

(1) Introduction

Line 49 and 57 – Some suggestions to change the wording. Comments were given in the manuscript attached herewith.

 R: Thanks for your insightful advice. Lines 49 and 57 in the manuscript have been changed.

(2) Methodology

Scientific names in the section 2.2 under the sub title “Sample collection”, have to be italicized.

Line 122, 123 and 149 – check whether, the elaborated symbols/terms have correctly mentioned. Some suggestions were given in the manuscript attached herewith.

 R: We have carefully corrected those issues (lines 96-109, lines 127-159) according to your suggestions.

(3) Results

Why table 1 has placed after the table 2? it is better if the tables are placed in order.

R: We apologize for the confusion and appreciate your remindier. We have changed the place of those two tables (lines 188-191, lines 212-215).

(4) Figure 4 – Does figure 4 give a meaningful relationship to support the argument related to it? If this relationship created with the 13C signatures and the tropic status (x axis 13C and y axis tropic level for two seasons) it would be more meaningful.

 R: Thanks for your suggestion. Based on your and the third referee’s comments, we have modified the figure 4. We analyzed the difference in proportional contribution of four food resources for fishes between two hydrological seasons by using non-metric multidimensional scaling (NMDS) (lines 163-166, lines 233-236, lines 239-240).

(5) Discussion

Authors have stated “Dongting derived carbon sources especially from benthic algae during the high-water period and relied more on seston and BOM in the dry season” in the discussion probably based on the values in the table 3 and 4. However, there is no strong evidences to prove this argument except the values in the table. If you can do a multivariate analysis for (i.e. PCA or MDS etc.) data in table 3 and 4, the positioning of diffident species in clusters during two seasons will give a good visualization of the changes during two seasons.

R: We appreciate your advice. To provide a better depiction of the changes, a multivariate analysis has been carried out. The result of NMDs supports the conclusion that proportional contributions of basal food sources to fish species exhibited much different between two hydrological seasons (lines 163-166, lines 233-236, lines 239-240).

(6) Note that in the result section, line 165 and 166, authors have stated that “No significant difference in the δ13C signatures of BOM between seasons was observed (F = 5.231, p = 0.062)”. It seems that these two arguments are conflicting.

R: We have depleted this sentence to make it more accurate (lines 174-175).

Specific comments.

(7) Line 49: Change “community” to “zone” or “area”. Meanwhile, change “Most” to “Most of the”.

R: We have changed them (line 52). Thank you for your suggestions.

(8) Line 57: “…….. facilitating the movement of potential source materials…..”. See the following suggestion to modify the sentence: it would be more convenience to the reader if you use "changes in potential food resources and movement of consumer taxa"

R: We have changed the sentence (line 60-61). Now it would be better.

(9) Line 91: scientific names have to be italicized

R: Thank you. We have italicized them (lines 96-109).

(10) Line 122: it is better to subscript "sample" in both places

R: We have changed them (lines 127-159) based on your suggestion.

(11) Line 122: It is more clear if you stated (Rsample/Rstandard) = 13C/12C or 15N/14N

R: Thanks for your suggestion. We have revised it (line 127) and apologize for our careless mistakes.

(12) Line 123: is the stated X (X=13C/12C or 15N/14N) is correct? Correct X should be 13C or 14C

R: Thank you. We have depleted that sentence (lines 127-128).

(13) Line 149: “Percent benthic”. is this term correct?

R: We have revised the formula (lines 151-159) according to your comment. The original formula was referred from Xu et al. (2007). The importance of littoral and benthic (or pelagic and littoral) resources to consumers (e.g. fish) was estimated using two-source mixing models. The current formula in the revised manuscript could be more precise.

Xu J., Zhang M., Xie P. 2007. Size-related shifts in reliance on benthic and pelagic food webs by lake anchovy. Ecoscience 14: 170-177.

(14) Line 221: “Relationship between trophic levels of fishes during the dry and wet seasons in Lake Dongting.” is this relationship a meaningful?

R: We apologize if our original figure 4 makes some confusion. We have changed this figure into multivariate analysis for NMDS (lines 234-235). Our original purpose was to test the seasonal difference in fish trophic levels from the community-level, because fish-specific data had been showed in figure 3. Perhaps it is meaningless to analyze the relationship of trophic levels between the dry and wet season.

(15) Line 269: “…….relied on more on seston and BOM in the dry season.” The statistical representation to come for this conclusion is not strong. if you can do a multivariate analysis for (PCA or a MDS etc.) data in table 3 an 4, the positioning of diffident species in clusters during two seasons will give a good visualization of the changes during two seasons.

R: We have conducted a multivariate analysis for NMDS based on your suggestion (lines 163-166, lines 233-236, lines 239-240). The result of NMDs supports the conclusion that proportional contributions of basal food sources to fish species exhibited much different between two hydrological seasons.

(16) Line 299: change “suffer smaller” to “low”.

R: We have changed it (line 356).

(17) Line 313: add “an”.

R: Thank you for your suggestion. We have rewritten discussion part according to Referee 3. We have changed the “an important carbon source” to “the primary energy source” (line 300).

Reviewer 4 Report

I suggest the following:

1. Please simplify the whole MS and consider the different structure of your paper.

2. Your data indicate individual species structure, which makes generalizations based upon means and more sophisticated stats difficult. 

3. Please examine your assumptions and see whether they influence your outcomes (especially calculation of trophic ranks in the food web).

Specific advice is given in the attached file.

No major problems. Spell check advised.

Author Response

Response to Reviewers’ Comments

Reviewer #4:

(1) Please simplify the whole MS and consider the different structure of your paper.

R: We thank the referee’s helpful comments. We have revised the manuscript's organization and simplified the entire MS.

(2) Your data indicate individual species structure, which makes generalizations based upon means and more sophisticated stats difficult.

R: We appreciate the valuable comments. We have tried our best to restructure and rewrite the result and discussion parts in previous draft to make it more concise and precise. Moreover, we have deleted some sentences to avoid confusion in our manuscript.

(3) Please examine your assumptions and see whether they influence your outcomes (especially calculation of trophic ranks in the food web).

R: Thank you for your valuable suggestion. We have carefully checked our study results, and restructured and rewritten the discussion and conclusion parts based on your and referee 2’ comments.

Specific advice is given in the attached file.

(4) Line 18: “….. in the wet season”. Needs further explanation. At present, non sequitur.

R: We have removed the passage and reworded the abstract (lines 17-18) according to the suggestion from referee 2.

(5) Line 18: “no uniform”. More explanation: as is, it nullifies your previous conclusion.

R: This sentence has been deleted (line 18) based on the comment of referee 2. Thank you.

(6) Line 20: Did they eat mainly other fish? What were the diet lists? Specific differences? Benthic algae vs other fish as food? Put more numerical data in the Abstract.

R: Thanks for your comment. We have added “likely” to make it more accurate (line 20). Evidences from stomach (gut) content data of fishes have not been quantified in this study, which can directly reveal an increase in carnivory during the dry season. We speculatively draw the conclusion from the perspectives of previous isotopic studies in Pantanal wetland (Wantzen et al., 2002) and Australia’s Wet Tropics rivers (Rayner et al., 2009, 2010). During the dry season fish become confined to the remaining body of water and food resources become limited, facilitating predatory-prey-interactions (Wantzen et al., 2002). Fish tend to feed on items from higher trophic levels, such as benthic and littoral invertebrates (Rayner et al., 2009).

Wantzen, K.M.; de Arruda Machado, F.; Voss, M.; Boriss, H.; Junk, W.J. 2002. Seasonal isotopic shifts in fish of the Pantanal wetland, Brazil. Aquatic Sciences 64, 239-251.

Rayner, T.S.; Pusey, B.J.; Pearson, R.G. 2009. Spatio-temporal dynamics of fish feeding in the lower Mulgrave River, north-eastern Queensland: the influence of seasonal flooding, instream productivity and invertebrate abundance. Marine and Freshwater Research 60, 97-111.

Rayner, T.S.; Pusey, B.J.; Pearson, R.G.; Godfrey, P.C. 2010. Food web dynamics in an Australian Wet Tropics river. Marine and Freshwater Research 61, 909-917.

(7) Line 26: What were the fluctuations?

R: We have deleted this sentence and reworded the abstract (lines 29-30). The term fluctuations is increasingly employed in hydrological research from river and floodplain systems. Most floodplain lakes, like Lake Dongting are subject to natural, mostly seasonal, fluctuations in water levels. The water level of Lake Dongting seasonally fluctuates over the annual hydrological cycle by precipitation patterns of the East Asian Monsoon zone.

(8) Line 115: change “power” to “powder”.

R: We’ve changed it (line 120). Thank you for your reminder.

(9) Line 117: “References for this chapter, please”.

R: We have added the reference (line 129).

(10) Line 150: With such a fish-specific data, it is not advisable to use means for these calculations across the board. More stratification advised.

R: We appreciate your helpful comment. The method we followed from Vander Zanden & Vadeboncoeur (2002) and other studies used the mean δ13C of consumer, and littoral and pelagic prey for calculations. Actually, we used individual data of each fish species to calculate the percentage contribution in this study. To clarify the method in our study, we have deleted “the mean” in lines 157-159.

Vander Zanden, M.J.; Vadeboncoeur, Y. 2002. Fishes as integrators of benthic and pelagic food webs in lakes. Ecology 83, 2152-2161.

Xu, J.; Zhang, M.; Xie, P. 2007. Size-related shifts in reliance on benthic and pelagic food webs by lake anchovy. Ecoscience 14, 170-177.

(11) Results part: Your results are very species-specific. I suggest to re-organize your results according to specific signatures per species. Now you have a tendency to lump and make generalizations partially shaped by your assumptions. In this way, you are close to circular argument(s).

R: We have rearranged the content of result part (lines 193-248). Thank you.

(12) Line 218: Figure 3. Explain abbreviations in the legend.

R: Abbreviations have been added (lines 228-231). Thank you for reminder.

(13) Line 221: Figure 4. Range of trophic levels during the dry season: 2.7-4.5; during the wet season 2.1-3.2. Means possibly greater intra- and inter-specific competition. Explore this please.

R: Figure 4 has been swapped out for another one according to referee 2. Our original figure 4 had been shown that Lake Dongting had longer food chain length and larger range of trophic levels of fishes during the low water period. The referee pointed out that it could mean possibly greater intra- and inter-specific competition. In floodplain systems, food resource availability was variable across hydrological seasons. During the wet season, fish appear to consume relatively more vegetative food and occupy lower trophic levels. However, fish tend to feed on items from higher trophic levels, such as benthic and littoral invertebrates during the dry season, reflecting consumption of a wide range of both detrital and benthic carbon sources. In that case trophic levels of fish was much higher.

(14) Discussion part: Discussion needs re-structuring and re-writing.

R: Our previous draft has undergone considerable revisions in response to your advice, and the discussion (lines 254–378) has been completely rewritten. We hope the revised manuscript could be acceptable for you. Thank you.

(15) Line 237: “Water-level fluctuations could affect the structure and functioning of the shallow lakes.” Which structure and which functioning? Be more specific please.

R: Thank you for your helpful comment. We have revised the sentence (lines 254-255).

(16) Line 302: “…….migration movements of fishes from the adjacent Yangtze River…”. It is not the alternative but an adjacent (parallel) factor.

R: Sorry for our carelessness. We have deleted that sentence (lines 357-360), and rewritten the discussion part according to your suggestion.

(17) Line 318: “ …….an important effect on the structure of East Lake Dongting…….”. You did not investigate "structure" as such (qualitative and quantitative species' changes and quantified dynamics of the food web of which they are part). Rather, you modeled their changing use of the available trophic resources.

R: Thanks for your suggestion. Indeed, community structure of East Lake Dongting has not been quantified in our study. So we have revised the sentence (lines 363-367) according to your comments.

(18) Line 326: “………untangle the complex trophic relationships in naturally fluctuating lakes…….”. You did not investigate "structure" as such (qualitative and quantitative species' changes and quantified dynamics of the food web of which they are part). Rather, you modeled their changing use of the available trophic resources.

R: Thanks for your suggestion. We wanted to investigate the trophic structure of Lake Dongting between the dry and wet seasons. To reduce the puzzle, we have reworded the conclusion (lines 375-378) to make it more precise and concise according to suggestions from you and referee 2.

Round 2

Reviewer 3 Report

Dear Authors,

Appreciate your prompt attempt of revising the manuscript according the comments. I would like to suggest come improvements for the title as follows

 "Influence of seasonal water level fluctuation on food web structure of a large floodplain lake in China”

Indicate what are the basal production sources? - line 153.

There is one suggestion for the table 3 and 4 (given in the line 222) and the line 255 256

Thank you

Author Response

Response to Reviewers’ Comments

Reviewer #3:

Dear Authors,

Appreciate your prompt attempt of revising the manuscript according the comments. I would like to suggest come improvements for the title as follows

"Influence of seasonal water level fluctuation on food web structure of a large floodplain lake in China”

Indicate what are the basal production sources? - line 153.

There is one suggestion for the table 3 and 4 (given in the line 222) and the line 255 256

Thank you

R: We appreciate the referee's kindly comments to us. We have revised those issues as your suggestions.

Specific comments:

(1) Title: suggestion for your consideration: "Influence of seasonal water level fluctuation on food web structure of a large floodplain lake in China.”

R: Thank you. Your suggestion has resulted in a change to the title (lines 2-3).

(2) Line 222: indicate why did you bold the numbers in the table 3 and 4.

R: Numbers in bold indicate assimilation of that source (1st-percentile values greater than 0). The explanation has been added to the title of Table 3 and Table 4 (lines 253-254, lines 256-257).

(3) Lines 255-256: take this sentence after the next one

R: The sentence has been change in order (lines 297-299). Thank you.

Reviewer 4 Report

Small corrections suggested. Minor English flaws which may be improved. See an attached file.

Generally good. Some very minor flaws.

Author Response

Response to Reviewers’ Comments

Reviewer #4:

Small corrections suggested. Minor English flaws which may be improved. See an attached file.

Specific comments.

(1) Line 158: Not obvious from Fig. 2 as a comparison is based on two different figures (a and b of Fig. 2). Refer rather to Tables 1-2.

R: We thank the referee for pointing out this reference to us. We have changed it (line 172).

(2) Line 235: changed into “as reflected”.

R: Thank you. We have changed it (line 271) as your helpful suggestion.

(3) Line 246: Also, possibly less water in various consumed tissues, hence elevated C13.

R: We appreciate the referee's kindly comments to us. Surely, as your suggested drought could enhance the 13C enrichment in plant tissues due to carbon isotopic discrimination. Therefore, most fish species were more enriched in δ13C when they largely consumed the 13C-enriched plant sources during the low water period. In our study, we found primary production sources assimilated by fishes were significantly varied in different hydrological seasons. That could be the most important reason why seasonal δ13C values shifted in fishes of Lake Dongting.

4) Line 260: change “large” into “broadly”.

R: Thank you. We’ve fixed the grammatical error (line 302).